# Lipid Biomarker Investigation of the Delivery and Preservation of Autochthonous Organic Carbon in the Pearl River and Its Contribution to the Carbon Sink: Evidence from the Water and Surface Sediment

**DOI:** 10.3390/ijerph192215392

**Published:** 2022-11-21

**Authors:** Mingxing Yang, Zaihua Liu, Hailong Sun, Min Zhao, Haibo He

**Affiliations:** 1Faculty of Resources and Environmental Engineering, Guizhou Institute of Technology, Guiyang 550003, China; 2State Key Laboratory of Environmental Geochemistry, Institute of Geochemistry, Chinese Academy of Sciences, Guiyang 550081, China

**Keywords:** biological carbon pump, autochthonous organic carbon, aquatic photosynthesis, carbonate weathering, karst carbon sink flux, pearl river basin

## Abstract

The molecular composition of the lipid biomarkers in the surface water, water column, and surface sediments collected along the Pearl River was investigated to identify the mechanisms of the delivery and preservation of autochthonous organic carbon (AOC) and to estimate its contribution to the carbon sink. The spatial distribution of these lipid biomarkers showed that samples collected at high-DIC-concentration sites (DIC: dissolved inorganic carbon) had prominent aquatic autochthonous signatures, while samples collected at low-DIC-concentration sites showed greater terrestrial contributions, which were described as the DIC fertilization effects. In the summer, typically, intense precipitation and flood erosion diluted the biogeochemical composition and carried terrestrial plant detritus. Therefore, the percentage of AOC (auto%) was higher in the winter than in the summer. According to the calculation of the lipid biomarkers, the values of the auto% were 65% (winter) and 54% (summer) in the surface water, 55.9% (winter) and 44.6% (summer) in the below-surface water, and 52.1% (winter) and 43.9% (summer) in the surface sediment, which demonstrated that AOC accounted for a major portion of the TOC. Vertical variability was mainly present in sites with intense flood erosion, which resulted in the mixing and deposition of resuspended sediments. There was a positive correlation of the clay content with the auto% value and the biogeochemical composition, showing that clay adsorbed the organic carbon in the water, vertically deposited it into the sediment, and was the dominant mechanism of the vertical delivery of organic carbon (OC). According to the new karst carbon sink model, based on coupled carbonate weathering and aquatic photosynthesis, the karst carbon sink flux (CSF) in the Pearl River was 2.69 × 10^6^ t/a which was 1.7 times the original estimation (1.58 × 10^6^ t/a), and this did not consider the formation of AOC. This indicated that previously, the contribution of the riverine system to the global karst carbon sink may have been highly underestimated.

## 1. Introduction

The biological carbon pump (BCP) is a process that converts atmospheric CO_2_ into recalcitrant organic carbon through photosynthesis and microbial action. Usually, the BCP can be divided into three distinct phases. The first phase is the fixation of atmospheric CO_2_ into dissolved inorganic carbon (DIC: CO_2_ (aq), HCO_3_^−^, and CO_3_^2−^), which occurs at the interface between the atmosphere and the surface of the water [1,2]. Following the first phase, phytoplankton in the aquatic system use the DIC during photosynthesis to make organic carbon (including lipids, carbohydrates, and proteins) [3,4,5,6]. Finally, organic carbon is transported into the ocean through flowing river water or is adsorbed by suspended particle matter and deposited into the sediment [7,8].

Due to the BCP effect, the OC transformed from DIC via aquatic photosynthesis is called autochthonous organic carbon and is derived from phytoplankton (mainly algae), which makes up a significant portion of the carbon stored in rivers [9,10]. On the other hand, the OC derived from the degradation products of higher plants is called allochthonous OC [11].

It is obvious that the BCP effect is an important sequestered carbon process. Especially in karst basins, rapid carbonate weathering can offer sufficient DIC as a carbon source for aquatic photosynthesis, and the amount of autochthonous OC can be considerable; therefore, its contribution to karst-related carbon sinks must be taken into account [12,13]. However, because autochthonous and allochthonous OCs are mixed together in the aquatic system, it is difficult to quantify the ratios of the two. Moreover, autochthonous OC is regarded as part of allochthonous OC, and the total amount of the carbon sink generated via carbonate weathering coupled with photosynthesis is greatly underestimated. Hence, for carbon sink assessment in the aquatic system, autochthonous and allochthonous OCs primarily need to be distinguished.

Another important factor in the carbon cycle of the riverine system is the vertical delivery and preservation of organic carbon, because more than 80% of it is buried in coastal marine systems adjacent to rivers [14,15,16,17]. Usually, OC is transported through inorganic clasts [18] or as discrete OC particles. Globally, the OC in suspended particles and sediment is thought to be largely associated with the clay content (<2 μm, %) [19]. Furthermore, OC may have a high affinity for clay surfaces rather than mineral surfaces [20,21,22,23]. Finally, clay particles can comprise a significant component of the TOC from the water column to the sediment [24]. Despite the global significance of the TOC delivery from large rivers to the ocean, the vertical diversity and amount of the TOC in these settings are still poorly understood.

Hence, the assessment of karst-related carbon sinks in the riverine system is still imprecise. There are two obstacles in addressing this problem: (1) material transportation in the riverine system is a complex process in terms of erosion, sediment, weathering, etc., which leads to difficulty in distinguishing the autochthonous and allochthonous OC; (2) the vertical delivery of OC is neglected, and, thus, the amount of carbon sink may be dramatically underestimated.

Lipid compounds (also known as molecular fossils) are derived from dead biological residues, which preserve the carbon skeleton of the original biochemical component and record the relevant information of the original biological parent material, and they can be used to identify their sources [25,26]. It has been reported that aquatic organisms usually produce short- to mid-chain n-alkanes. Inversely, terrigenous higher plants usually produce long-chain n-alkanes [27,28,29]. Monounsaturated fatty acids (MUFAs) (e.g., C16:1ω and C18:1ω) are used to represent organic carbons derived from fresh algal [30]. Polyunsaturated fatty acids (PUFAs) (e.g., C18:2ω and C18:3ω) are used to represent organic carbons derived from higher plant [31]. In addition, sitosterol (29^Δ5^) and campesterol (28^Δ5^) sterols are biomarkers of terrestrial plants, while epibrassicasterol (28^Δ5,22^) is found in phytoplankton in particular [32]. Another phytoplanktonic marker, 24-methylenecholesterol (28^Δ5,24^), is characteristic of diatoms [32]. Therefore, lipid biomarkers were widely used to distinguish OC sources [33,34].

The contribution of riverine water to the carbon sink generated via carbonate weathering coupled with aquatic photosynthesis has been examined in previous studies. However, concerns regarding the vertical delivery and preservation of the AOC in big rivers have not been fully understood. It is highly important to take big rivers into account when carbon balance is researched at the global level. The Pearl River, the third largest river in China, is characterized by its typical carbonate cover and the large amount of its sediment load, offering a perfect location for the study of the carbon sink generated via carbonate weathering coupled with photosynthesis. In this study, we followed the seasonal and spatial transportation and conversion of DIC and OC during the BCP process with chemical and lipid biomarkers and a phytoplankton analysis in both the water column and sediments of the Pearl River to achieve the following: (1) to describe the seasonal variations of the geochemical and lipid biomarkers in both the water column and sediments in the Pearl River; (2) to examine the vertical distribution of autochthonous OC based on the BCP effect; and (3) to estimate the amount of autochthonous OC in the Pearl River.

## 2. Materials and Methods

### 2.1. Study Area and Sampling Sites

#### 2.1.1. Study Area

The Pearl River originates in the Wumeng Mountains in the western Yunnan Province, flows east, and empties into the South China Sea [35]. Xijiang, Beijiang, and Dongjiang are the main tributaries of the Pearl River (Figure 1). Xijiang is the largest tributary and is located in the upper and middle reaches of the Pearl River basin [36]. The upper reach includes two large tributaries: Nanpanjiang and Beipanjiang (first-order tributaries of Xijiang). The middle reach includes the large tributaries of Liujiang, Guijiang, and Yujiang (first-order tributaries of Xijiang). The lower reach includes the Beijiang and Dongjiang tributaries (Figure 1). The Pearl River basin is dominated by a subtropical to tropical monsoon climate. From east to west, the mean annual temperature and precipitation across the basin ranges from 14℃ to 22℃ and 1200 mm to 2200 mm, respectively [37].

In the entire basin, the annual average runoff and sediment load of Pearl River from 1954 and 2010 were 283 × 10^9^ m^3^ and 72.4 × 10^6^ t, respectively [38]. From April to September, the accumulative runoff and sediment load accounted for 84% and 94% of the annual totals. During the flood season, the runoff and sediment load accounted for 50% and 72% of the annual totals [39]. In geographical and geological terms, the Pearl River basin is characterized with a wide distribution of carbonate rocks (especially in the upper and middle reaches, which account for approximately 45% of the basin as shown in Figure 1). In addition, clastic rock and granite are mainly located in the upper reach (approximately 27% of the basin) and lower reach (approximately 16% of the basin), respectively [40].

#### 2.1.2. Sample Collection

Carbon sinks produced via carbonate weathering coupled with aquatic photosynthesis appeared in both the river water and sediments. During this process, the amount and mechanism of carbon sinks were quite different in the surface water, water column, and sediment. In order to show the difference, we collected water samples at 5 cm below the water surface in winter (dry season) and summer (wet season). All of the sampling locations were representative of the Pearl River and its main distributaries in terms of their lithological, climatic, and hydrological impacts (Table 1). Water samples from the water column and surface sediment were collected in six sections (XJ2, XJ3, XJ4, XJ6, BJ, and DJ) at five depths (0 m, 2 m, 4 m, 6 m, and 8 m) to research the vertical distribution. Among those sections, XJ2, XJ3, XJ4, and XJ6 represented the Xijiang River, and BJ and DJ represented the Beijiang River and Dongjiang River, respectively.

Here, 500 mL of water was collected in a brown bottle at each site for GC–MS analysis (types and concentrations of OC). To another bottle of 500 mL of water, 1% MgCO_3_ and Lugol’s solution were added to determine the Chlorophyll a (Chl a) levels and biomass. Sediment samples (0~5 cm of the riverbed) from six sections (shown in Table 1) were taken.

### 2.2. Chemical Analysis

A Merck field test box was applied to measure the HCO_3_^−^ and Ca^2+^ levels in the water samples [41]. Anion and cation concentrations were measured via ion chromatography (ICS–90, Dionex) and ICP–OES (Vista MPX, Varian, USA), respectively. The water temperature (T), pH, dissolved oxygen (DO), and electrical conductivity (EC) of the water samples were measured in situ using the WTW Technology MultiLine 350i. All the meters were calibrated before use, and the standard error was less than 2%. The resolutions of the pH, T, EC, and DO measurements were 0.01, 0.01 °C, 0.01 μs cm^−1^, and 0.01 mg/L, respectively. The pCO_2_ (CO_2_ partial pressure) and SIc (calcite saturation index) were calculated using the WATSPECT program [42]. A blood counting chamber was used to determine the amount of biomass in the water samples. A UV spectrophotometer was used to measure the OD680 and turbidity of the water samples according to the handbook of monitoring and analysis methods of water and wastewater [43].

Chl a was extracted from water samples by using a 90% acetone solution before analysis. After supersonic treatment, the extracts were assessed with a UV spectrophotometer, and the Chl a concentration was determined according to standard methods established by the Ministry of Environmental Protection of China [43]. Before the in situ monitoring process, all of the monitoring instruments were calibrated to ensure the accuracy of the monitoring data.

### 2.3. Biomarker Analysis

MeOH and hexane were used to extract the organic compounds in water samples. All the extracted substances were then condensed to 1 mL using the rotary evaporator. A portion of each extract was saponified using 1 M KOH in aqueous Methanol-MeOH. Neutral and acidic lipids were extracted into hexane from the saponified samples. Fatty acids were converted to methyl esters using BF_3_–MeOH. Both fatty acids and neutral lipids were separated from other lipid classes via silica gel chromatography. Sterols were derivatized to trimethylsilyl ethers using BSTFA in acetonitrile and were heated to 70 °C for 30 min [9]. The lipid biomarkers in the sediment samples (3.0 g dry weight) were extracted using a Soxhlet apparatus with 210 mL of a mixture of dichloromethane and methanol (2:1, v:v) for 20 h. Similar subsequent processes were used to evaluate the water samples.

Gas chromatography (GC) (Agilent 7890 A) was used to measure fatty acids, sterols, and n-alkanes. The size of the column was 30 m × 0.32 mm × 0.25 μm (item: DB5). In the qualitative analysis of the organic compounds, the retention times of the external standards were compared (Sigma–Aldrich Company, St. Louis, MO, USA).

Meanwhile, the concentrations of the compounds were measured by calculating their total ion current (TIC) peak area. The GC-MS running program for each type of compound was: (1) Fatty acids: the initial temperature was 60 °C, which then increased to 150 °C at the rate of 40 °C/min. Then, it increased from 150 °C to 240 °C at the rate of 3 °C/min, which was maintained for 15 min. (2) Sterols: the temperature was raised from 80 °C (maintained for 1 min) to 200 °C at the rate of 25 °C/min, and it was then raised from 200 °C to 250 °C at the rate of 3 °C/min. It was subsequently increased to 300 °C (rate: 1.8 °C/min, maintained for 2 min). (3) n-alkanes: the temperature was maintained at 70 °C for 1 min and then raised to 140 °C at the rate of 10 °C/min, and it was subsequently increased by 3 °C/min to 310 °C, which was maintained for 15 min [44].

### 2.4. Grain Size of Suspended Sediment and TSS Analysis

All surface sediment and core samples were pretreated with 10–20 mL of 30% H_2_O_2_ to remove organic matter, washed with 10% HCl to remove carbonates and mollusk fragments, rinsed with deionized water, and then placed in an ultrasonic bath for several minutes to facilitate dispersion. Then, a Malvern Mastersizer 2000 grain size analyzer was used to measure the grain size of both the sediments and TSS (the accuracy of this method was 1 mg/L).

## 3. Results

### 3.1. Seasonal Variation in Chemical, Phytoplankton, and Biomarker Compositions and Their Implications for Carbonate Weathering Coupled with Aquatic Photosynthesis

#### 3.1.1. Seasonal Distribution of Biochemical Compositions in Surface Water and the DIC Fertilization Effect

(1) Chemical composition

The chemical composition of the water during the winter (dry season) and summer (wet season) in terms of HCO_3_^−^, Ca^2+^, pCO_2_, pH, and EC is shown in Figure 2. HCO_3_^−^ made up the main portion of DIC; its spatial and seasonal distribution in the Pearl River was quite varied. Overall, sites with a primarily carbonate bedrock offered an abundant DIC source. XJ1, for example, had DIC concentrations of 4.6 mmol/L and 3.5 mmol/L in the winter and summer, respectively, whereas the lowest value appeared in the silicate area, GJ1 (upper Guijiang), with DIC concentrations of 0.4 mmol/L and 0.5 mmol/L. There was a great difference in the HCO_3_^−^ concentrations between the karst area and the non-karst area. The same patterns were found in the Ca^2+^ concentrations. More specifically, the Ca^2+^ concentrations ranged from 88.0 mg/L (XJ1) to 10.0 mg/L (GJ1) in the winter (average 46.9 ± 22.6 mg/L) and 60.0 mg/L (XJ1) to 8.0 mg/L (GJ1) in the summer (average 33.9 ± 16.1 mg/L).

The pCO_2_ value ranged from 4.1 ppm to 2985.0 ppm (average 1684.0 ± 928.1 ppm) in the winter and 0.4 ppm to 3235.9 ppm (average 1503.9 ± 1170.2 ppm) in the summer. The SIc value ranged from −1.73 to 1.52 (average −0.07 ± 0.97) in the winter and −1.72 to 1.41 (average 0.17 ± 0.08) in the summer.

(2) Phytoplankton composition

Due to the BCP effect, DIC (mainly HCO_3_^−^) is utilized during photosynthesis in aquatic ecosystems. The products include OC and phytoplankton, especially algae. Therefore, the features of phytoplankton and their relationship with the DIC distribution were of great importance for understanding the mechanism of the BCP effect. The average concentration of Chl a in the winter and summer was 6.3 ± 1.6 mg/m^3^ and 8.6 ± 2.1 mg/m^3^, respectively, showing that photosynthesis was more intense in the summer than in the winter [45]. In addition to in situ aquatic photosynthesis, terrestrial flushing materials could also carry Chl a into inland water.

Another product of photosynthesis in aquatic systems is the accumulation of biomass (mainly phytoplankton, such as algae). The average phytoplankton biomass was 2.0 × 10^6^ ± 0.8 × 10^6^ cell/L and 1.7 × 10^6^ ± 0.7 × 10^6^ cell/L in the winter and summer, respectively. The highest phytoplankton biomass (3.5 × 10^6^ cell/L in the winter and 2.9 × 10^6^ cell/L in the summer) was detected at XJ3 (where the DIC concentration was high, water flow was low, and the concentration of TSS was low), in which large amounts of DIC and a stable environment enabled photosynthesis. Inversely, with low DIC and high water flow, photosynthesis was weaker at DJ, which caused a low biomass (1.0 × 10^6^ cell/L in the winter and 0.9 × 10^6^ cell/L in the summer). The OD680 values exhibited the same trend with biomass. These statistical correlations illustrated that the phytoplankton biomass was more intense at the upstream sites (with high DIC concentrations). However, at the downstream sites (with low DIC concentrations), the phytoplankton biomass also slightly increased in the aquatic system (Figure 3).

#### 3.1.2. Biomarker Distribution in Surface Water—Implications for Carbonate Weathering Coupled with Aquatic Photosynthesis

The lipid biomarkers, in terms of fatty acids, sterols, and n-alkanes, in surface water were measured and are listed in Table 2 to demonstrate their implications for source appointment between aquatic and terrigenous organic carbon.

(1) Fatty acids

In the winter, the concentrations of the total fatty acids (TFA = SSFA + BSFA + MUFA + PUFA) ranged from 11.4 μg/L to 311.8 μg/L with an average of 75.9 ± 7.8 μg/L, while, in the summer, they ranged from 126.8 μg/L to 689.1 μg/L with an average of 255.6 ± 132.3 μg/L. The highest and lowest values were found in XJ2 and GJ1, respectively. The TFA concentrations varied in the HCO_3_^−^ concentrations, ranging from 129.7 μg/L (ranging from 62.7 μg/L to 311.8 μg/L) in the winter and 488.1 μg/L (ranging from 316.2 μg/L to 689.1 μg/L) in the summer at sites with high HCO_3_^−^ concentrations to 35.1 μg/L (ranging from 11.4 μg/L to 56.1 μg/L) in the winter and 219.6 μg/L (ranging from 126.8 μg/L to 406.9 μg/L) in the summer at sites with low HCO_3_^−^ concentrations. Saturated fatty acids were divided into straight and branched saturated fatty acids (SSFAs and BSFAs). SSFAs, the most abundant class, predominated over long-chain saturated fatty acids, accounting for 63.8% (winter) and 59.6% (summer) of the TFA. Unsaturated fatty acids were divided into MUFAs and PUFAs, which can indicate the diatom growth process in the aquatic systems and which have been used to determine autochthonous OC [46]. The mean percentages of MUFA were 13.2% (winter) and 11.8% (summer), whereas those of PUFA and BSFA were 13.1% (winter) and 11.8% (summer) and 9.9% (winter) and 14.1% (summer) of the TFA, respectively (Table 2). C16:0 (palmitic acid) was the most abundant compound, which accounted for an average of 20.0% ± 8.3% of the TFA. C16:1ω (palmitoleic acid) was the major MUFA and was used to indicate autochthonous production. Water samples at XJ3 contained the most abundant C16:1ω (23.5 μg/L), whereas DJ had the lowest value (3.6 μg/L). In the summer, the concentrations of C16:1ω at XJ3 and DJ were 19.8 μg/L and 6.2 μg/L, respectively, illustrating the diversity of the different sites. The presence of C16:1ω indicated that the photosynthesis of phytoplankton was intense in the Pearl River [47]. Inversely, fluvial erosion in the river carried the degradation products of terrestrial plants, for example, C18:2ω (linoleic acid), hence indicating an allochthonous source. The concentrations of C18:2ω was between 0.2 μg/L and 17.9 μg/L in the winter and was between 8.9 μg/L and 41.1 μg/L in the summer.

(2) Sterols

The concentrations of total sterols ranged from 1.1 μg/L to 3.6 μg/L (average: 2.4 ± 0.9 μg/L, winter) and from 2.6 μg/L to 5.1 μg/L (average: 3.8 ± 1.6 μg/L, summer). 28^Δ5,22^ was the most abundant sterol at all sites, which varied from 0.3 μg/L to 1.7μg/L (winter) and from 0.2 μg/L to 1.3μg/L (summer). The highest concentration of 28^Δ5,22^ was detected in the upstream major tributary (XJ3) which was mainly dominated by carbonate rocks (Figure 1). In contrast, the lowest concentration was observed at DJ which was dominated by silicate rock (Figure 1). Cholest-5-en-3β-ol (27^Δ5^) had the same distribution pattern as 28^Δ5,22^, and its average values were 0.8 ± 0.3 μg/L (winter) and 0.5 ± 0.1 μg/L (summer). 28^Δ5^ and 29^Δ5,22^ were biomarkers for sources of terrestrial plants [48]. Their concentrations in the Pearl River ranged from 0.2 μg/L to 0.8 μg/L (winter) and 0.6 μg/L to 1.9 μg/L (summer) and from 0.1 μg/L to 0.4 μg/L (winter) and 0.5 μg/L to 2.1 μg/L (summer), respectively. Contrary to 28^Δ5,22^ and 27^Δ5^, 28^Δ5^ and 29^Δ5,22^ were found to be enriched at downstream sites where various materials were drained.

(3) n-Alkanes

In the winter, the concentrations of n-alkanes were between 601.3 μg/L and 2316.5 μg/L with an average of 1311.8 ± 516.5 μg/L. Meanwhile, in the summer, these concentrations ranged from 881.6 μg/L to 2115.2 μg/L with an average of 1522.1 ± 631.2 μg/L. The carbon chain numbers of n-alkanes were from C12 to C34, and the most abundant compound was C17 (The concentration of C17 was 35.6 μg/L to 311.2 μg/L in the winter with an average of 149.3 ± 58.6 μg/L and 28.2 μg/L to 278.5 μg/L in the summer with an average of 115.8 ± 49.5 μg/L). The values of CPI_HC_ were 2.99 ± 1.03 and 3.49 ± 1.06 on average and ranged from 1.18 to 4.99 (winter) and from 1.61 to 5.06 (summer) (Table 2). Upstream tributaries showed the highest value of CPI_HC_. TAR_HC_ represents the terrestrial-to-aquatic ratio which was used to evaluate the relative proportions of autochthonous and allochthonous hydrocarbons in the aquatic system [49]. The value of TAR_HC_ in the water sample was 0.26 to 0.63 with an average value of 0.52 ± 0.11 in the winter, and 0.29 to 0.81 with an average value of 0.58 ± 0.13 in the summer, respectively.

By combining the distribution of the biochemical compositions and lipid biomarkers, the regression analysis method was used to obtain the correlation and determination coefficients between the parameters. The *p* values for the trends were also listed. A positive correlation was found between the DIC concentration and the growth of aquatic biomass, autochthonous ratio, and specific parameters (C16:1ω/C18:2ω), which revealed that a high DIC concentration in an aquatic system could offer more carbon for the photosynthesis process, hence deriving more phytoplankton biomass and forming more autochthonous OC. This phenomenon was reported as the DIC fertilization effect (Figure 4).

### 3.2. Spatial Patterns of Biomarkers in the Water Column and Surface Sediments as a Consequence of BCP Effects

#### 3.2.1. Distribution of Lipid Biomarkers in the Water Column and Surface Sediments

Table 3 shows the lipid biomarkers of the water column and surface sediment samples from the six main sites in the Pearl River. Compared with the surface water conditions, the vertical distribution also showed the same seasonal patterns as those discussed above. Variations were mainly derived from the complex processes that occurred during the formation and transportation of autochthonous and allochthonous OC.

(1) Fatty acids

Similar to those from the surface water, the water column samples from XJ3 contained the highest TFA values with concentrations that ranged from 363.2 μg/L to 451.6 μg/L. There was little vertical change because XJ3 was characterized by a low flow speed (a lacustrine condition). Hence, the fluctuation was weak. In contrast, XJ2 and XJ6 showed intense flood erosion features, leading to higher concentrations at the surface. In addition, the water levels at these sites were low; intense erosion can result in the resuspension of the bottom sediment and vertical change.

Straight saturated fatty acids (SSFAs) made up the major portion of the TFA in the water column samples, making up 39.8 to 72.5% (winter) and 36.5 to 68.2% (summer) of the TFA. This showed the dominant aspect of aquatic OC input. Branched saturated fatty acids (BSFAs) and PUFAs exhibited lower contributions with values ranging from 12 to 22% and from 21 to 26% of the TFA, respectively.

(2) Sterols

The sterols distribution in the water column samples were mainly 28^Δ5,22^ and 27^Δ5^ (representing an aquatic autochthonous organic source) and 28^Δ5^ and 29^Δ5,22^ (representing a terrigenous allochthonous organic source). The concentration of 28^Δ5,22^ ranged from 0.2 to 1.7 mg/L in the water column samples and from 16.2 to 142.7 mg/kg in the sediment samples. The highest and lowest values appeared at XJ4 and DJ, respectively. In contrast, 29^Δ5^ was mainly detected at XJ6 with the highest value of 46.8 mg/kg in the summer, revealing the increased input of terrestrial plant detritus caused by intense flood erosion. The ratio of 27^Δ5^/29^Δ5,22^ + 29^Δ5^ was positively correlated with the input of the autochthonous OC source, and the results also showed that values in winter were higher than in summer.

(3) n-Alkanes

The water column and surface sediment samples in the Pearl River showed a wide distribution range of quantifiable n-C12 to n-C35 alkanes, which revealed a mixed aquatic and terrestrial source. Algal and photosynthetic products were generally characterized by low, odd-numbered-carbon n-alkanes (n-C17), which were predominant in the Pearl River water and sediments. Similarly, the long-chain homologues (n-C27, n-C29, and n-C31) derived from terrestrial higher-plant waxes and the odd-numbered, mid-chain n-alkanes (n-C23 and n-C25) derived from lower plants were mainly detected in the summer. Other n-alkane parameters, such as TAR_CH_, ranged from 0.21 to 0.81 (average 0.59 ± 0.18) in the water column samples and from 0.46 to 0.89 (average 0.65 ± 0.11) in the surface sediment samples. The lowest value appeared at XJ3 (0.21 in the winter and 0.22 in the summer) when the flow was slow and photosynthesis was intense, resulting in the accumulation of autochthonous OC.

In summary, sites with intense flood erosion characteristics, for example, XJ2 and XJ6, demonstrated vertical variation because of the complex process, in terms of flood erosion and photosynthesis, on the surface and the resuspended sediments at the bottom. First, the sun supplies the energy for photosynthesis, which enhances the surface synthesis of autochthonous OC. Therefore, the values at all stations were higher than those at the bottom. Second, flood erosion carries terrestrial plant detritus into the river and results in increasing allochthonous OC. In contrast, in samples from XJ3, BJ, and DJ, the distribution of the lipid biomarkers showed little variation from the surface to the bottom because of their weak flow.

#### 3.2.2. Grain Size Distribution and Its Influence on the BCP Effects

The grain size of suspended sediment can reflect the vertical transportation of suspended sediment [50]. The available data demonstrate considerable variation in the particle size characteristics of sediment from different sites in response to variations in source materials and other physiographic controls [51].

We analyzed the vertical distribution of the suspended sediments (0, 2, 4, 6, and 8 m from the water surface) and the surface sediments at the six stations (Table 4 and Figure 5). It is clear that a river with a high flow speed can cause various grain sizes to be mixed with characteristic coarse sand. XJ2, for example, was located at the main stream of the Xijiang River and exhibited obvious river erosion features with the d(0.5) (median grain size) value ranging from 101.16 to 157.07 μm in the winter and 128.33 to 165.21 μm in the summer. On the other hand, the water flow at XJ3 was relatively slow, resulting in the accumulation of fine sand. The grain size at XJ3 ranged from 6.07 to 7.61 μm in the winter and 7.61 to 7.89 μm in the summer. In terms of seasonal influence, in the summer, there was usually more precipitation, which caused intense flood erosion, and, thus, the grain size was bigger in the summer than in the winter. This phenomenon was quite obvious at XJ2, XJ4, and XJ6, which were characterized by high flow rates, resulting in more coarse sand and a tremendous seasonal difference between the winter and summer. In contract, XJ3, BJ, and DJ had low flow velocities; therefore, the sediments were mainly composed of fine sand, and the seasonal difference was not so obvious. In addition, by comparing XJ2 and XJ3, it is clear that intense erosion also generated the resuspension of the surface sediment and showed that the grain size of the bottom water was higher than the value of the surface water. Within the XJ3 section, grain sizes of all the vertical water samples were quite small and were almost the same, reflecting a lacustrine feature.

In Figure 6, it was demonstrated that the main mechanism of vertical transportation during the BCP process was derived by the adsorption of clay material, which allowed the formed OC at the surface area to vertically move to the bottom and finally be stored as sediment. Hence, when estimating the carbon sink in the aquatic system, the OC buried in the sediment must be taken into account.

### 3.3. Carbon Sink Estimation Based on Photosynthesis Coupled with Carbonate Weathering

The source identification of organic carbon in the Pearl River is very difficult because of the mixed effects of rock weathering, photosynthesis, and stream erosion. Lipid biomarkers can provide unique bioinformation and are extensively used to distinguish autochthonous and allochthonous OC. The ratio of autochthonous and allochthonous OC represents the in situ productivity of the aquatic system in the allochthonous input.

The results showed that the autochthonous organic source in the surface water of the Pearl River was the predominant input, with averages of 65% and 54% in the winter and summer, respectively (Figure 7). This value in the water column was 55.9% (winter) and 44.6% (summer), and, in the surface sediment, it was 52.1% (winter) and 43.9% (summer) (Figure 8). The results indicated that the autochthonous OC in the aquatic system was mainly derived from the in situ primary productivity.

It is obvious that, due to the BCP effects, the OC derived from the aquatic photosynthetic use of carbonate-weathering-derived DIC was vertically transported through adsorption by clay materials and was finally stored in the sediment. According to the rock-weathering-related carbon sink model, the pure karst carbon sink in the riverine system was composed of three parts: (1) the dissolved inorganic carbon in the water from carbonate weathering (half of the DIC came from atmospheric CO_2_ that was dissolved in water, and the other half came from the carbonate rock); (2) the autochthonous organic carbon derived from photosynthesis in the aquatic system; and (3) the autochthonous organic carbon in the sediments. Hence, the carbon sink flux (CSF) in the karst river could be calculated with the following equation [12]: CSF = Q_water_ × (DIC/2 + TOC_water_ × Auto%_water_) + F_sediment_ × TOC_sediment_ × Auto%_sediment_, where Q_water_, F_sediment_, TOC_water_, TOC_sediment_, Auto%_water_, and Auto%_sediment_ represent the flow rate, sediment flux, and TOC concentration in the water and sediment and the ratio of autochthonous to TOC in the water and sediment, respectively. Wei et al. reported that the average annual TOC concentration in the Pearl River was 4.71 mg/L, and the DIC concentration was 11.17 mg/L [52]. The average concentration of the TOC in sediment was 11,000 mg/kg, the annual flow rate was 283 × 10^9^ m^3^, and the sediment flux was 72.4 × 10^6^ t (Ministry of Water Resource, PRC, 1954–2010) [38].

By taking the average value of the autochthonous to TOC ratio in both the water column and surface sediments in the Pearl River and by using the above values, we estimated that the karst carbon sink in the Pearl River was 2.69 × 10^6^ t/a which was 1.7 times the original estimation (CSF = Q × DIC/2 = 1.58 × 10^6^ t/a). Hence, the contribution of the riverine ecosystem to the global karst carbon sink may be larger than we ever previously thought as we did not consider the BCP effect.

## 4. Conclusions

In this study, the winter (dry season, January to February) and summer (wet season, July to August) water column samples and sediment samples from six stations in the Pearl River were collected to examine their lipid biomarkers and biochemical compositions. In addition, the full range of sediment grain sizes was also determined to reveal the mechanism of the BCP effects in the riverine system and to estimate the autochthonous OC and its contribution to the carbon sink. The results showed the following: (1) Both the concentrations of OC and the biochemical compositions showed significant spatial and seasonal variations, which were the combined results of the processes of aquatic photosynthesis, carbonate weathering, flood erosion, particle adsorption, and suspended-sediment deposition. More specifically, sites with high DIC concentrations in the water can result in a high autochthonous OC ratio. In the summer, there is usually intense precipitation and flood erosion which dilutes the biogeochemical composition and carries terrestrial plant detritus; therefore, the auto% value was higher in the winter than in the summer. According to the calculation of the lipid biomarkers, the auto% values were 65% (winter) and 54% (summer) in the surface water, 55.9% (winter) and 44.6% (summer) in the water column, and 52.1% (winter) and 43.9% (summer) in the surface sediment, which demonstrated that autochthonous OC accounted for the main portion of the TOC. (2) Vertical variability mainly appeared at sites with intense flood erosion characteristics. There was a positive correlation between the clay content and the auto% value and biogeochemical composition, showing that the adsorption of organic carbon in the water by clay and the vertical deposition into the sediment was the dominant mechanism of the vertical delivery of OC. (3) According to the new carbon sink model based on carbonate weathering coupled with aquatic photosynthesis, the karst carbon sink flux (CSF) in the Pearl River was 2.69 × 10^6^ t/a which was 1.7 times the original estimation (1.58 × 10^6^ t/a) and which did not consider the BCP effect, revealing that the contribution of the riverine ecosystem to the global carbon sink may be larger than we ever previously thought. These results showed that due to the high DIC content in karst rivers, a large carbon sink can be produced through aquatic photosynthesis and sedimentation. Hence, future research and applications of carbon neutrality should pay more attention to the coupled carbonate weathering and aquatic photosynthesis processes. In addition, artificial methods (land-use regulation, soil improvement, aquatic-algae cultivation, etc.) that can increase the DIC concentration in water or convert DIC into OC need to be explored to enhance karst carbon sinks.

## Figures and Tables

**Figure 1 ijerph-19-15392-f001:**
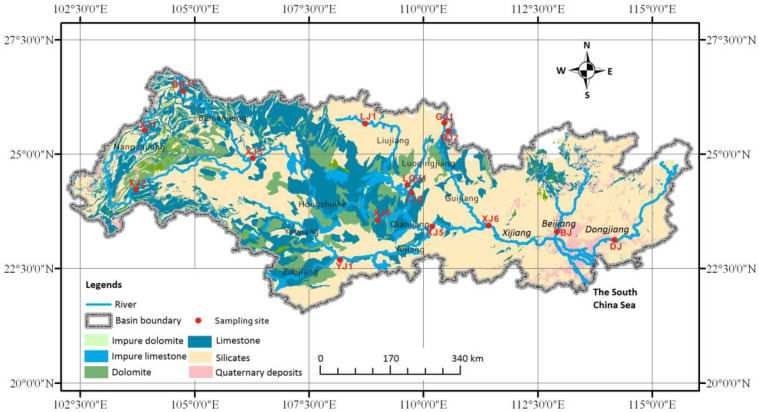
Geology of the Pearl River and sampling sites.

**Figure 2 ijerph-19-15392-f002:**
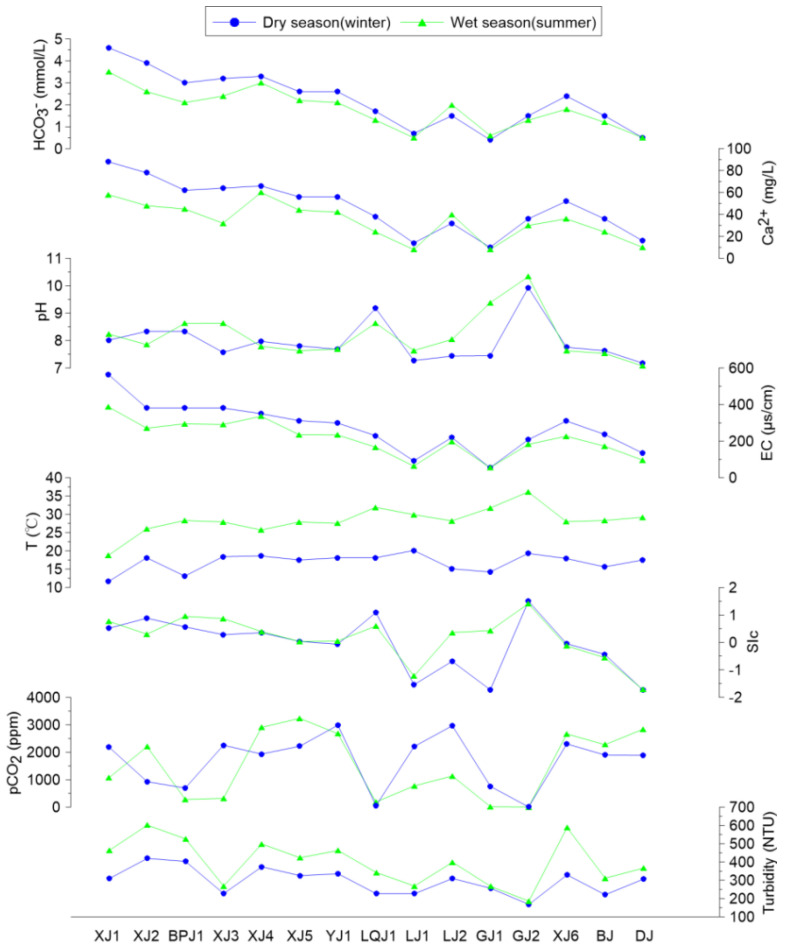
Geochemical parameters of water samples in the Pearl River basin.

**Figure 3 ijerph-19-15392-f003:**
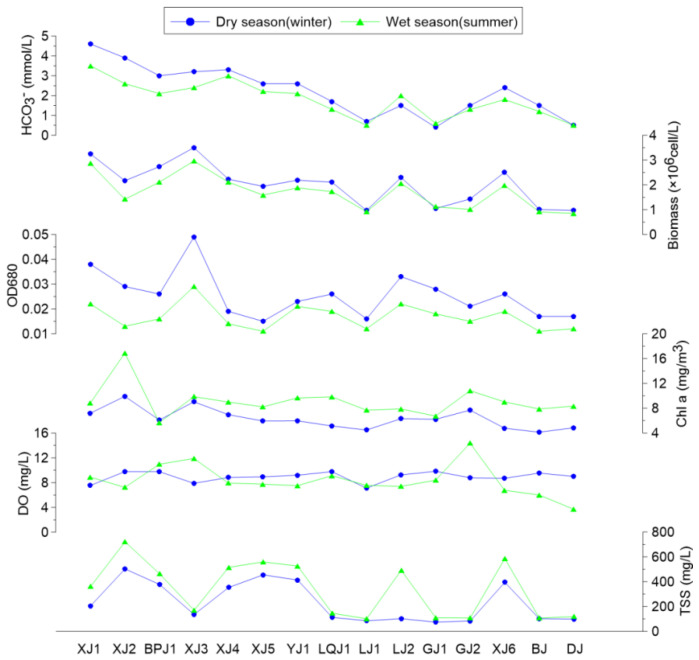
Characteristics of phytoplankton in water samples in the Pearl River basin.

**Figure 4 ijerph-19-15392-f004:**
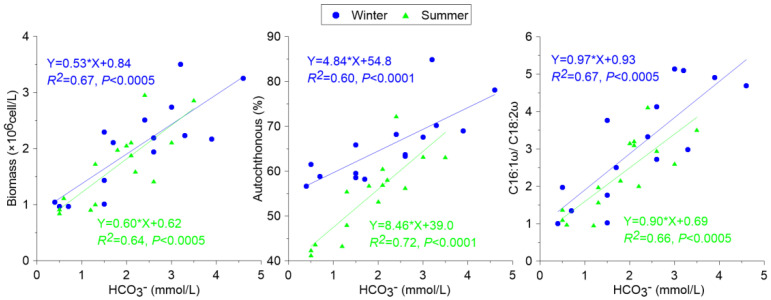
Relationship between photosynthesis and carbon weathering.

**Figure 5 ijerph-19-15392-f005:**
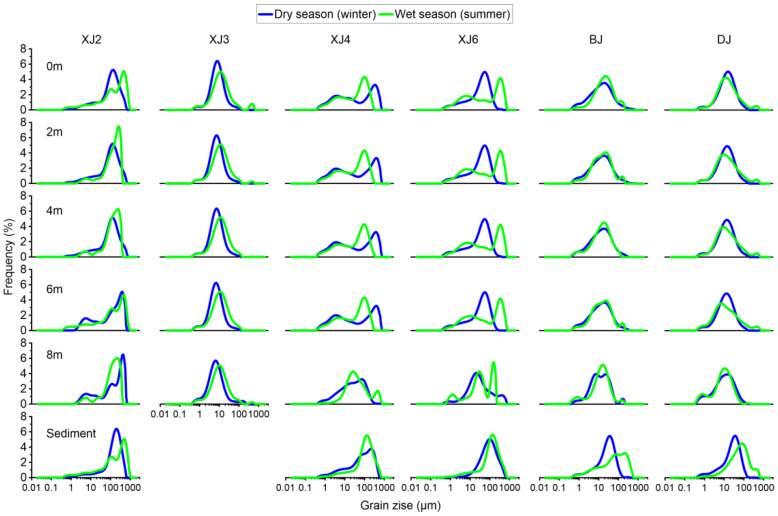
Grain size distribution of suspended particles and surface sediment.

**Figure 6 ijerph-19-15392-f006:**
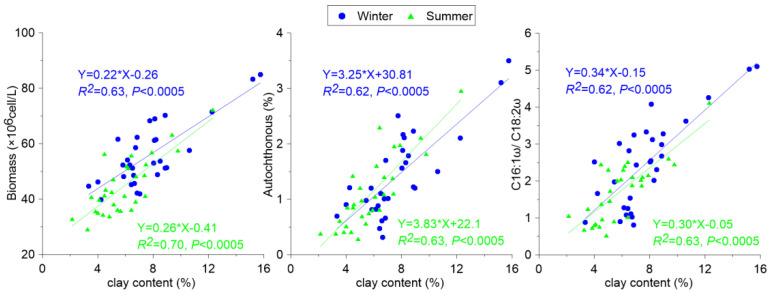
Relationship between photosynthesis and sediment grain size.

**Figure 7 ijerph-19-15392-f007:**
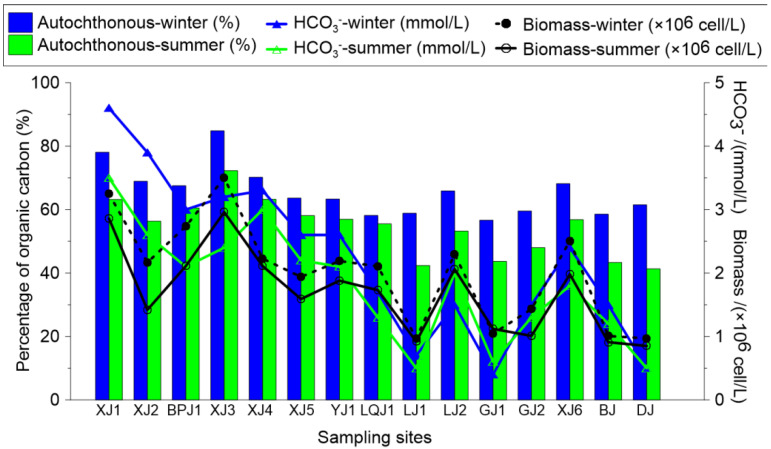
Seasonal and spatial distributions of autochthonous (%) and allochthonous (%) OC and the hydrochemical (HCO_3_^−^) and biological (C16:1ω/C18:2ω and biomass) parameters; TAR_FA_ = (C24 + C26 + C28)/(C12 + C14 + C16); autochthonous (%) OC = [ΣSCFA(C12~C18)/TAR_FA_] × 100%; and allochthonous (%) OC = [ΣLCFA(C22~C30)/TAR_FA_] × 100%.

**Figure 8 ijerph-19-15392-f008:**
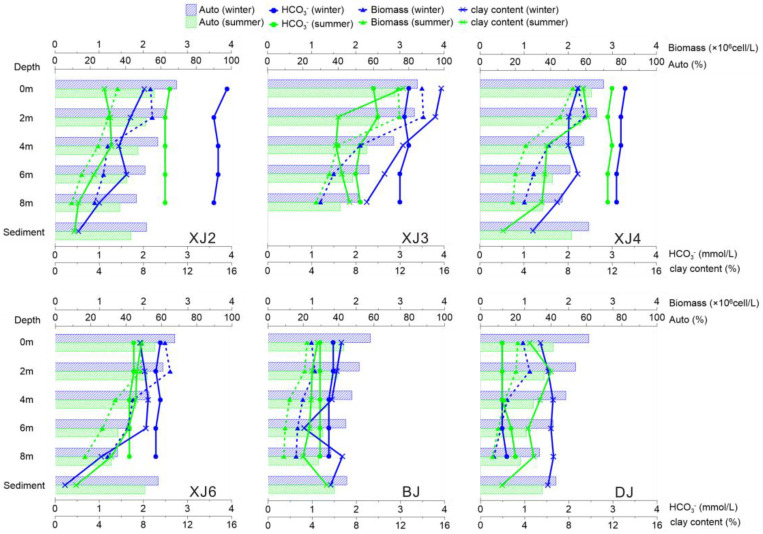
Vertical variation in autochthonous ratio of organic carbon and relevant biogeochemical indexes in water column and surface sediments of Pearl River.

**Table 1 ijerph-19-15392-t001:** Description of sampling sites.

Site	Location	Tributary Description	Sample Types			TSS * (mg/L)	T * (°C)	Precipitation * (mm/a)
			Surface Water	Column Water	Sediment			
XJ1	25°36′18″ N; 103°49′33″ E	Upstream of Nanpanjiang	√			1290	18.6	1000.0
XJ2	24°01′13″ N; 103°36′20″ E	Midstream of Nanpanjiang	√	√	√	680	20.2	963.7
BPJ1	26°29′59″ N; 103°44′07″ E	Upstream of Beipanjiang	√			2610	18.7	1127.6
XJ3	24°57′47″ N; 106°08′55″ E	Junction of Nanpanjiang and Beipanjiang	√	√	√	170	19.1	921.1
XJ4	23°44′04″ N; 109°13′43″ E	Downstream of Hongshuihe	√	√	√	628	20.8	1499.8
YJ1	22°48′38″ N; 108°18′46″ E	Junction of Zuojiang and Youjiang	√			241	21.6	1304.2
LQJ1	24°24′09″ N; 109°36′25″ E	Downstream of Luoqingjiang	√			132	20.0	1257.7
LJ1	25°13′07″ N; 109°23′47″ E	Upstream of Liujiang	√			143	19.0	1493.0
LJ2	24°14′36″ N; 109°43′13″ E	Junction of Liujiang and Luoqingjiang	√			132	21.6	1304.2
XJ5	23°27′50″ N; 110°09′39″ E	Junction of Qianjiang and Yujiang	√			350	21.4	1726.7
GJ1	25°31′17″ N; 110°11′41″ E	Upstream of Guijiang	√			67	18.0	2052.0
GJ2	25°06′19″ N; 110°24′55″ E	Upstream of Guijiang	√			92	18.7	1926.0
XJ6	23°28′40″ N; 111°17′11″ E	Downstream of Xijiang	√	√	√	311	21.1	1503.6
BJ	23°41′21″ N; 113°03′26″ E	Downstream of Beijiang	√	√	√	129	20.7	1900.1
DJ	23°09′28″ N; 114°16′12″ E	Downstream of Dongjiang	√	√	√	107	22.1	1821.2

* Annual mean value.

**Table 2 ijerph-19-15392-t002:** Biomarkers at various sites in Pearl River and their relevant values.

	Fatty Acids				Sterols			n-Alkanes		
Site	SSFA (%)	BSFA (%)	MUFA (%)	PUFA (%)	28^Δ5,22^ (μg/L)	29^Δ5^ (μg/L)	27^Δ5^/29^Δ5,22^ +29^Δ5^	TAR_HC_	CPI_HC_	C17 (μg/L)
XJ1	72.0/63.2	2.2/9.6	19.7/10.6	6.1/16.6	1.2/0.7	0.2/0.4	4.79/3.56	0.38/0.46	1.56/2.01	296.3/208.6
XJ2	66.5/49.6	8.3/18.2	17.5/12.6	7.7/19.6	1.5/0.8	0.4/1.1	4.32/3.12	0.61/0.72	2.09/3.06	255.3/158.8
BPJ1	60.3/52.2	8.4/11.5	16.5/13.9	14.7/22.4	0.9/0.6	0.2/0.5	3.53/3.11	0.55/0.61	2.12/2.49	266.3/188.5
XJ3	61.2/58.3	4.1/5.1	23.8/21.6	10.8/15.0	1.7/1.5	0.1/0.2	7.29/6.89	0.26/0.29	1.18/1.61	311.2/278.5
XJ4	65.4/55.6	3.8/14.7	19.2/15.2	11.6/14.5	0.9/0.6	0.2/0.6	3.90/3.01	0.47/0.51	2.92/4.03	216.3/156.6
YJ1	53.5/49.0	10.8/22.1	13.7/10.8	22.1/24.7	1.5/1.1	0.2/0.7	1.89/1.68	0.59/0.61	3.43/3.98	243.2/146.3
LQJ1	62.9/64.8	12.7/16.8	6.6/5.1	17.8/13.3	1.1/0.6	0.3/0.4	2.79/2.81	0.61/0.73	3.79/4.28	163.2/112.6
LJ1	55.0/58.9	21.7/24.8	10.0/5.5	13.4/10.8	0.4/0.4	0.2/0.8	1.45/1.22	0.48/0.41	2.28/3.56	35.6/31.1
LJ2	77.1/73.9	3.7/9.7	14.8/15.3	4.4/1.1	0.4/0.5	0.2/0.4	0.81/0.89	0.52/0.58	4.99/5.01	35.6/28.2
XJ5	62.2/64.9	4.8/8.9	12.8/14.2	20.1/12.0	0.6/0.6	0.3/0.3	2.02/2.88	0.52/0.66	2.88/2.58	189.0/136.1
GJ1	62.7/58.2	17.2/19.5	8.7/13.2	11.4/9.1	0.4/0.4	0.2/0.4	0.67/0.56	0.49/0.51	3.32/2.96	47.7/61.9
GJ2	63.3/70.1	16.5/15.8	7.2/9.8	13.0/4.3	0.7/0.9	0.2/0.4	1.59/1.41	0.46/0.55	3.52/4.12	69.9/72.8
XJ6	58.6/53.3	12.2/15.2	15.6/8.5	13.6/23.0	1.3/0.6	0.3/0.5	1.99/1.58	0.51/0.62	2.33/2.80	68.0/50.1
BJ	72.5/68.2	6.4/10.2	11.7/11.3	9.5/10.3	0.9/0.5	0.4/0.5	0.89/0.93	0.68/0.75	4.71/5.06	88.8/68.3
DJ	63.9/53.6	10.1/16.0	9.9/10.1	16.0/20.5	0.3/0.4	0.4/0.6	1.11/1.21	0.72/0.81	3.16/4.88	45.2/39.8
mean	63.8 ± 6.1/59.6 ± 7.39	9.5 ± 5.5/14.1 ± 5.2	13.9 ± 4.8/11.8 ± 3.9	12.8 ± 4.8/14.5 ± 6.6	0.9 ± 0.4/0.7 ± 0.3	0.3 ± 0.1/0.5 ± 0.2	2.60 ± 1.7/2.32 ± 1.5	0.52 ± 0.11/0.58 ± 0.13	2.99 ± 1.03/3.49 ± 1.06	149.3 ± 100.2115.8 ± 71.7

Data before and after “/” represent winter and summer results, respectively; SSFA represents straight saturated fatty acids; BSFA represents branched saturated fatty acids; TAR_HC_ = (C27 + C29 + C31)/(C15 + C17 + C19); CPI_HC_ = 0.5[(C25 + C27 + C29 + C31 + C33/C24 + C26 + C28 + C30 + C32) + (C25 + C27 + C29 + C31 + C33/C26 + C28 + C30 + C32 + C34)]; C17 represents heptadecane.

**Table 3 ijerph-19-15392-t003:** Lipid biomarker distribution in water column and surface sediment of Pearl River in seasons of winter (W) and summer (S), respectively.

Site	Depth (m)	Fatty Acids								Sterols						n-Alkanes					
		SSFA (%)		BSFA (%)		MUFA (%)		PUFA (%)		28^Δ5,22^ (μg/L)		29^Δ5^ (μg/L)		27^Δ5^/29^Δ5,22^ +29^Δ5^		TAR_HC_		CPI_HC_		C17 (μg/L)	
		W	S	W	S	W	S	W	S	W	S	W	S	W	S	W	S	W	S	W	S
XJ2	0	66.5	49.6	8.3	18.2	17.5	12.6	7.7	19.6	1.5	0.8	0.4	1.1	4.32	3.12	0.61	0.72	2.09	3.06	255.3	158.8
	2	64.8	47.1	9.9	19.3	16.7	13.2	8.6	20.4	1.6	0.7	0.5	1	4.21	3.01	0.63	0.73	2.11	3.05	231.5	129.6
	4	62.6	45.8	8.1	20.1	18.8	12.1	10.5	22.0	1.3	0.6	0.4	0.9	4.24	2.86	0.69	0.74	2.32	3.09	201.6	115.7
	6	60.2	42.2	9.3	23.2	17.6	11.8	12.9	22.8	1.1	0.7	0.6	0.8	4.01	2.41	0.71	0.78	2.27	3.11	168.2	114.2
	8	48.8	41.9	16.1	22.8	19.8	12.8	15.3	22.5	1.1	0.4	0.4	0.8	3.89	2.62	0.73	0.76	2.14	3.16	155.7	109.8
	S	53.6	38.2	15.8	23.6	13.8	16.1	16.8	22.1	32.5	21.3	15.4	24.1	4.18	2.97	0.62	0.68	2.13	2.25	171.2	121.8
XJ3	0	61.2	61.2	4.1	5.1	23.8	21.6	10.8	15.0	1.7	1.5	0.1	0.2	7.29	6.89	0.26	0.29	1.18	1.61	311.2	278.5
	2	61.6	58.9	5.3	5.4	22.6	20.8	10.5	14.9	1.7	1.5	0.1	0.2	7.31	6.88	0.26	0.27	1.13	1.65	289.6	267.3
	4	60.8	58.3	6.8	6.4	23.1	19.5	9.3	15.8	1.6	1.3	0.1	0.3	7.11	6.51	0.23	0.22	1.23	1.71	257.1	254.9
	6	60.1	57.2	6.5	7.6	23.8	17.6	9.6	17.6	1.5	1.5	0.2	0.2	7.01	6.37	0.21	0.31	1.18	1.66	261	239.1
	8	58.2	56.9	7.9	8.9	24.1	22.1	9.8	12.1	1.4	1.1	0.1	0.2	6.99	6.68	0.28	0.23	1.14	1.58	245.8	211.3
	S *																				
XJ4	0	65.4	55.6	3.8	14.7	19.2	15.2	11.6	14.5	0.9	0.6	0.2	0.6	3.9	3.01	0.47	0.51	2.92	4.03	216.3	156.6
	2	61.3	57.3	4.3	12.6	22.2	14.9	12.2	15.2	0.9	0.6	0.2	0.6	3.81	2.89	0.45	0.55	2.95	3.77	215.8	138.9
	4	58.2	51.9	7.7	16.2	15.3	16.6	18.8	15.3	0.8	0.5	0.3	0.6	3.63	2.71	0.51	0.52	2.91	3.62	208.1	166.8
	6	51.6	45.8	9.3	21.1	19.8	18.2	19.3	14.9	0.8	0.6	0.5	0.5	4.01	2.77	0.46	0.46	2.87	3.78	201.6	127.5
	8	42.1	38.9	8.8	24.6	24.5	20.2	24.6	16.3	0.7	0.4	0.3	0.4	3.87	2.58	0.45	0.48	2.61	4.11	186.2	132.9
	S	45.3	40.6	11.9	21.9	21.0	19.3	21.8	18.2	142.7	116.2	24.5	22.1	6.43	6.12	0.46	0.55	2.77	3.87	267.9	251.5
XJ6	0	58.6	53.3	12.2	15.2	15.6	8.5	13.6	23.0	1.3	0.6	0.3	0.5	1.99	1.58	0.51	0.62	2.33	2.8	68	50.1
	2	56.2	48.6	12.9	17.8	15.1	9.9	15.8	23.7	1.3	0.5	0.4	0.5	2.01	1.49	0.55	0.66	2.35	2.76	63.2	46.3
	4	48.7	46.9	13.6	18.4	20.5	8.6	17.2	26.1	1.2	0.4	0.2	0.3	1.96	1.38	0.57	0.73	2.41	2.84	56.9	45.1
	6	42.1	40.2	14.1	17.3	24.5	14.4	19.3	28.1	1.1	0.3	0.3	0.4	1.86	1.39	0.62	0.81	2.26	2.66	60.1	41.6
	8	39.8	36.5	14.3	19.8	24.4	18.2	21.5	25.5	0.9	0.5	0.3	0.4	1.73	1.42	0.63	0.73	2.18	2.51	45.6	39.5
	S	46.9	45.2	15.5	19.0	19.8	14.7	17.8	21.1	86.3	72.1	31.3	46.8	2.05	1.65	0.54	0.66	2.17	2.89	167.3	105.2
BJ	0	72.5	68.2	6.4	10.2	11.7	11.3	9.5	10.3	0.9	0.5	0.4	0.5	0.89	0.93	0.68	0.75	4.71	5.06	88.8	68.3
	2	65.6	67.2	8.7	11.5	14.5	9.8	11.2	11.5	0.9	0.5	0.4	0.4	0.87	0.92	0.69	0.73	4.71	5.01	86.4	66.2
	4	66.7	66.1	8.4	12.6	11.4	9.0	13.5	12.3	0.8	0.6	0.7	0.4	0.91	0.96	0.68	0.78	4.74	4.71	82.1	61.8
	6	68.2	64.2	6.2	13.1	13.8	10.8	11.8	11.9	0.5	0.4	0.3	0.6	0.84	0.91	0.66	0.81	4.81	4.75	78.9	55.6
	8	63.5	63.8	11.3	14.1	12.9	9.3	12.3	12.8	0.7	0.7	0.5	0.4	0.91	0.89	0.71	0.63	4.55	4.68	68.1	46.3
	S	62.8	61.3	8.8	15.1	15.2	12.0	13.2	11.6	25.1	19.8	13.8	22.9	0.93	0.81	0.65	0.68	4.36	4.71	31.6	26.7
DJ	0	63.9	53.6	10.1	16.0	9.9	10.1	16.0	20.5	0.3	0.4	0.4	0.6	1.11	1.21	0.72	0.81	3.16	4.88	45.2	39.8
	2	62.5	52.5	8.9	14.1	11.3	11.7	17.3	21.7	0.3	0.2	0.4	0.5	1.12	1.14	0.72	0.75	3.17	4.89	46.7	38.1
	4	62.1	51.3	9.2	11.9	10.5	13.2	18.2	23.6	0.2	0.3	0.3	0.4	1.05	1.12	0.73	0.79	3.22	4.85	44.2	36.3
	6	60.1	52.9	9.9	13.5	12.8	11.5	17.2	22.1	0.4	0.4	0.3	0.6	1.03	1.15	0.77	0.74	3.19	4.66	42.8	37.8
	8	58.9	52.7	11.0	9.9	13.2	12.8	16.9	24.6	0.4	0.3	0.3	0.5	1.02	1.04	0.72	0.73	3.12	4.81	43.6	39.6
	S	63.7	55.8	8.3	7.6	12.2	14.1	15.8	22.5	21.4	16.2	16.9	24.8	0.85	0.99	0.74	0.89	3.64	4.53	41.2	36.1

* Sediment at XJ3 was not collected.

**Table 4 ijerph-19-15392-t004:** Seasonal comparison of chemical parameters of water column at various sites in Pearl River.

Site	Depth (m)	d(0.5) (μm)		DO (mg/L)		pH		EC (μs/cm)		T (℃)	
		Winter	Summer	Winter	Summer	Winter	Summer	Winter	Summer	Winter	Summer
XJ2	0	108.58	165.21	9.77	7.60	8.34	8.08	381	428	18.1	23.4
	2	102.58	150.26	9.14	8.41	7.99	8.06	313	435	17.8	23.6
	4	101.16	132.66	9.05	8.22	7.96	8.01	311	431	17.8	21.3
	6	131.36	129.63	8.92	7.69	7.89	7.90	309	425	17.8	23.9
	8	157.07	128.33	8.76	7.41	7.80	7.82	306	418	17.6	23.8
XJ3	0	7.61	11.35	7.89	10.27	7.57	8.56	381	320	18.4	28.9
	2	6.93	10.78	7.61	11.98	7.93	8.58	402	329	18.6	28.3
	4	6.86	10.77	7.69	9.54	7.88	8.57	382	322	18.1	27.9
	6	6.35	10.27	7.87	8.16	7.87	8.16	383	326	18.0	26.6
	8	6.07	9.46	7.86	8.03	7.88	8.02	381	329	18.0	26.7
XJ4	0	29.62	42.22	8.83	7.92	7.97	7.79	350	336	18.6	25.7
	2	29.38	41.52	8.98	7.61	7.95	7.86	353	335	18.6	26.3
	4	29.20	41.42	8.62	7.67	7.95	7.78	349	337	18.7	25.8
	6	28.48	40.98	8.55	7.33	7.94	7.83	348	336	18.6	26.0
	8	24.26	33.38	8.61	7.30	7.58	7.89	349	336	18.7	26.5
XJ6	0	36.08	49.02	8.70	6.77	7.76	7.63	310	227	17.9	28.0
	2	35.56	48.22	8.75	6.46	7.80	7.88	312	172.4	17.8	28.6
	4	35.42	47.05	8.73	6.80	7.79	7.85	312	174.7	17.8	28.7
	6	35.20	44.15	8.71	6.99	7.79	7.86	312	167.4	17.6	28.9
	8	23.91	31.07	8.70	6.93	7.78	7.81	312	167.9	17.6	29.1
BJ	0	13.37	18.59	9.54	5.72	7.63	7.07	237	96.2	15.6	29.2
	2	12.82	14.25	10.37	5.65	7.89	7.54	241	96.6	15.0	29.8
	4	12.70	13.85	10.27	5.54	7.87	7.69	239	93.6	14.8	31.5
	6	12.43	13.69	10.33	5.19	7.85	7.33	240	94.0	14.8	30.0
	8	10.83	12.69	10.33	4.54	7.92	7.47	240	104.6	14.9	28.6
DJ	0	13.78	13.69	9.00	6.01	7.12	7.52	135	172.6	17.5	28.3
	2	12.39	12.10	9.36	6.26	7.15	7.63	138	171.8	17.5	29.8
	4	11.99	12.08	9.28	5.79	7.34	7.58	136	168.1	17.1	29.6
	6	11.48	10.27	9.23	5.75	7.48	7.52	136	174.4	17.1	28.7
	8	9.73	8.737	9.15	5.70	7.43	7.48	135	170.6	17.3	29.4

d(0.5): median grain size.

## Data Availability

Not applicable.

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
