# Peer review of "Lipid Biomarker Investigation of the Delivery and Preservation of Autochthonous Organic Carbon in the Pearl River and Its Contribution to the Carbon Sink: Evidence from the Water and Surface Sediment"

_ijerph, 2022, doi:10.3390/ijerph192215392_

Round 1
Reviewer 1 Report
Dear Authors
I believed the paper is worth to be published in Int. J. Environ. Res. Public Health. However, some corrections are also needed.
1. Details about Data Analysis should bed added. Any software package has been used? If, yes, solver setting and solution method should be presented.
2. Please add the "assumptions" which were considered in this work (both experiments and theoretical).
3. It is required to perform the sensitivity analysis.
4. The text of the abstract and the conclusion are very similar to each other. It is suggested to rewrite the conclusion briefly.
5. The future prospects should be addressed with consideration of the opportunities and challenges
Sincerely yours,
Author Response
Dear Reviewer,
We would like to express our sincere appreciation for your careful reading and helpful comments. Those comments are all valuable and very helpful for revising and improving our paper, as well as the important guiding significance to our researchers. We have carefully read all comments and have tried our best to revise the manuscript as per reviewers suggestions, which we hope to meet with acceptance requirements. In the following are the responses to the reviewer’s comments.
First of all, the reviewer required English editing. We have polished the language with the help of Mdpi English Editing. Please check the revised manuscript and the certification for details.
Reviewer comments: I believed the paper is worth to be published in Int. J. Environ. Res. Public Health. However, some corrections are also needed.
- Details about Data Analysis should bed added. Any software package has been used? If, yes, solver setting and solution method should be presented.
Re: Thank you for your suggestion. We used regression analysis to illustrate the relationship between different characteristics. Please check the revised manuscript for details.
- Please add the "assumptions" which were considered in this work (both experiments and theoretical).
Re: Thank you for your suggestion. The theory and basic steps of coupled carbonate weathering with aquatic photosynthesis carbon sink has been described in the introduction. Please check the revised manuscript for details.
- It is required to perform the sensitivity analysis.
Re: Thank you for your suggestion. The sensitivity analysis has been added and highlighted in red in the revised manuscript. Please check the revised manuscript for details.
- The text of the abstract and the conclusion are very similar to each other. It is suggested to rewrite the conclusion briefly.
Re: Thank you for your suggestion. The conclusion has been modified and highlighted in red in the revised manuscript. Please check the revised manuscript for details.
- The future prospects should be addressed with consideration of the opportunities and challenges.
Re: Thank you for your suggestion. The consideration and opportunities of carbon sink model based on coupled carbonate weathering with aquatic photosynthesis have been addressed and highlighted in red in the revised manuscript. Please check the revised manuscript for details.
That’s all for now. Hope you’re satisfied with our replies to your concerns.
With best wishes,
Mingxing Yang
Reviewer 2 Report
See attached file.

Author Response
Dear Reviewer,
We would like to express our sincere appreciation for your careful reading and helpful comments. Those comments are all valuable and very helpful for revising and improving our paper, as well as the important guiding significance to our researchers. We have carefully read all comments and have tried our best to revise the manuscript as per reviewers suggestions, which we hope to meet with acceptance requirements. In the following are the responses to the reviewer’s comments.
First of all, the reviewer required English editing. We have polished the language with the help of Mdpi English Editing. Please check the revised manuscript and the certification for details.
Reviewer comments:
- 1. Comments have been highlighted in attached file (manuscript).
Re: Respected reviewer thank you again for highlighting the misunderstandings in our manuscript. According to your suggestion, the formatting and spelling errors are corrected and highlighted in red in the revised manuscript. Please check the revised manuscript for details.
- 2. Page 14 / rows 419-436: perhaps an additional methodological material explaining how the respective equations were arrived at would be welcome.
Re: Thank you for your suggestion, the equations are explained in the revised manuscript.
- 3. Page 15 / rows 438-440: “The present study examines the lipid biomarker and biochemical composition in both the water surface / water column and surface sediments of the Pearl River collected along the river and 439 at six depth....
Re: Thank you for your suggestion, this part of conclusion has been modified and highlighted in red in the revised manuscript. Please check the revised manuscript for details.
That’s all for now. Hope you’re satisfied with our replies to your concerns.
With best wishes,
Mingxing Yang
Reviewer 3 Report
My comments are about the samples collection: please specify the difference between 5 cm depth and o, 2, 4, 6 and 8 m depth samples in a better way, it is unclear. Line 159: replace “MeOH” with “Methanol-MeOH”. Line 163: use the subscript for the number 3 of “BF3”. Line 166: no comma between “3.0 g” and “dry weight”. Line 187: what is the accuracy of the method?
English editing is required.

Author Response
Dear Reviewer,
We would like to express our sincere appreciation for your careful reading and helpful comments. Those comments are all valuable and very helpful for revising and improving our paper, as well as the important guiding significance to our researchers. We have carefully read all comments and have tried our best to revise the manuscript as per reviewers suggestions, which we hope to meet with acceptance requirements. In the following are the responses to the reviewer’s comments.
First of all, the reviewer required English editing. We have polished the language with the help of Mdpi English Editing. Please check the revised manuscript and the certification for details.
Reviewer comments:
- 1. My comments are about the samples collection: please specify the difference between 5 cm depth and 0, 2, 4, 6 and 8 m depth samples in a better way, it is unclear.
Re: Thank you for your suggestion, we have modified the description the sampling aims and differences, are addressed in the text:
“The carbon sink produced by coupled carbonate weathering with aquatic photosynthesis appeared in both the river water and sediments. During this process, the amount and mechanism of carbon sink were quite different from the surface water, water column and sediment”.
”Water samples from the water column and surface sediment were collected in six sections (XJ2, XJ3, XJ4, XJ6, BJ and DJ) at five depths (0 m, 2 m, 4 m, 6 m and 8 m) to research the vertical distribution”.
- 2. Line 159: replace “MeOH” with “Methanol-MeOH”.
Re: Thank you for your suggestion, this has now been corrected.
- 3. Line 163: use the subscript for the number 3 of “BF3”.
Re: Thank you for your suggestion, this has now been corrected.
- 4. Line 166: no comma between “3.0 g” and “dry weight”.
Re: Thank you for your suggestion, this has now been corrected.
- 5. Line 187: what is the accuracy of the method?
Re: Thank you for your suggestion, we have described the accuracy the method.
That’s all for now. Hope you’re satisfied with our replies to your concerns.
With best wishes,
Mingxing Yang
